# Comprehensive Genetic Analyses of Inherited Peripheral Neuropathies in Japan: Making Early Diagnosis Possible

**DOI:** 10.3390/biomedicines10071546

**Published:** 2022-06-29

**Authors:** Masahiro Ando, Yujiro Higuchi, Junhui Yuan, Akiko Yoshimura, Takaki Taniguchi, Fumikazu Kojima, Yutaka Noguchi, Takahiro Hobara, Mika Takeuchi, Jun Takei, Yu Hiramatsu, Yusuke Sakiyama, Akihiro Hashiguchi, Yuji Okamoto, Jun Mitsui, Hiroyuki Ishiura, Shoji Tsuji, Hiroshi Takashima

**Affiliations:** 1Department of Neurology and Geriatrics, Kagoshima University Graduate School of Medical and Dental Sciences, Kagoshima 890-8520, Japan; k3251170@kadai.jp (M.A.); higuchi0302@m2.kufm.kagoshima-u.ac.jp (Y.H.); jhyuans@gmail.com (J.Y.); yoshiaki@m2.kufm.kagoshima-u.ac.jp (A.Y.); taniguchi.neurology@gmail.com (T.T.); fumikazukls@yahoo.co.jp (F.K.); k4869223@kadai.jp (Y.N.); script13starmine@gmail.com (T.H.); k2970270@kadai.jp (M.T.); juntakeijun@gmail.com (J.T.); k1614558@kadai.jp (Y.H.); sacky@m3.kufm.kagoshima-u.ac.jp (Y.S.); aki-hagu@m.kufm.kagoshima-u.ac.jp (A.H.); kamoto@m.kufm.kagoshima-u.ac.jp (Y.O.); 2Department of Physical Therapy, School of Health Sciences, Faculty of Medicine, Kagoshima University, Kagoshima 890-8520, Japan; 3Department of Neurology, Graduate School of Medicine, The University of Tokyo, Tokyo 113-8655, Japan; mituij-tky@umin.ac.jp (J.M.); ishiura@m.u-tokyo.ac.jp (H.I.); tsuji@m.u-tokyo.ac.jp (S.T.); 4Institute of Medical Genomics, International University of Health and Welfare, Chiba 107-8402, Japan

**Keywords:** inherited peripheral neuropathies (IPNs), Charcot–Marie–Tooth (CMT), gene panel sequencings, *RFC1*

## Abstract

Various genomic variants were linked to inherited peripheral neuropathies (IPNs), including large duplication/deletion and repeat expansion, making genetic diagnosis challenging. This large case series aimed to identify the genetic characteristics of Japanese patients with IPNs. We collected data on 2695 IPN cases throughout Japan, in which *PMP22* copy number variation (CNV) was pre-excluded. Genetic analyses were performed using DNA microarrays, next-generation sequencing-based gene panel sequencing, whole-exome sequencing, CNV analysis, and *RFC1* repeat expansion analysis. The overall diagnostic rate and the genetic spectrum of patients were summarized. We identified 909 cases with suspected IPNs, pathogenic or likely pathogenic variants. The most common causative genes were *MFN2*, *GJB1*, *MPZ*, and *MME*. *MFN2* was the most common cause for early-onset patients, whereas *GJB1* and *MPZ* were the leading causes of middle-onset and late-onset patients, respectively. Meanwhile, *GJB1* and *MFN2* were leading causes for demyelinating and axonal subtypes, respectively. Additionally, we identified CNVs in *MPZ* and *GJB1* genes and *RFC1* repeat expansions. Comprehensive genetic analyses explicitly demonstrated the genetic basis of our IPN case series. A further understanding of the clinical characteristics of IPN and genetic spectrum would assist in developing efficient genetic testing strategies and facilitate early diagnosis.

## 1. Introduction

Inherited peripheral neuropathies (IPNs) are a complex group of peripheral nervous system diseases with a wide range of phenotypic and genotypic diversity. Charcot–Marie–Tooth disease (CMT) is the most common type of IPN, which commonly presents with progressive sensory-motor or motor neuropathy, foot deformity, and altered tendon reflexes. CMT is classified by the motor nerve conduction velocity (MNCV) and inheritance pattern. Generally, CMT can be classified as demyelinating type (median MNCV of <35 m/s), axonal type (median MNCV of >45 m/s), and intermediate type (median MNCV of 35–45 m/s) [1]. The motor-dominant and sensory-dominant IPN phenotypes are further classified into distal hereditary motor neuropathy and hereditary sensory neuropathy, respectively. Additionally, IPN encompasses hereditary sensory autonomic neuropathy, when both sensory and autonomic nervous systems are involved, and hereditary neuropathy with liability to pressure palsy.

To date, >140 genes have been linked to IPNs (Neuromuscular Home Page. Available online: https://neuromuscular.wustl.edu/ (accessed on 1 June 2022)) [2]. IPN diagnosis has been largely improved in the last decade by next-generation sequencing (NGS) technology development, which allows gene panel sequencing that simultaneously targets many genes. Recently, developed and experimentally validated bioinformatics tools, such as CovCopCan, have further improved the genetic diagnosis of copy number variations (CNVs) in IPN-related genes [3]. Furthermore, short tandem repeat expansions were associated with various neurological disorders, including peripheral nervous system involvement. Recessive intronic pentanucleotide repeat AAGGG expansions of *replication factor complex subunit 1* (*RFC1*) were demonstrated as a genetic basis of cerebellar ataxia, sensory neuropathy, and vestibular areflexia syndrome (CANVAS) in 2019 [4]. Thereafter, the clinical spectrum of *RFC1*-related disorders was extended, including, but not limited to, pure sensory or sensory-dominant neuropathy and motor neuronopathy [5,6,7]. Thus, *RFC1* analysis becomes essential for IPN diagnosis.

Constant and continuous genetic study improvements in this study enable us to portray their genetic and clinical characteristics based on our monocenter collection of approximately 2700 Japanese patients with IPNs. This is the premise of making early IPN diagnosis possible.

## 2. Materials and Methods

### 2.1. Patients

We collected 2695 unrelated Japanese IPNs patients from neurological and neuropediatric departments throughout Japan from 2007 to March 2022. The clinical diagnosis of IPN was made by each neurologist or pediatric neurologist. Among these patients, family history or consanguinity were found positive from 954 cases, and 1698 cases were sporadic; 45 cases had no available family history data. All patients with the demyelinating type were confirmed negative for *Peripheral Myelin Protein 22* (*PMP22*) duplication/deletion using fluorescence in situ hybridization (FISH) or multiplex ligation probe amplification (MLPA). Genomic deoxyribonucleic acid (DNA) was extracted from the peripheral blood or the saliva using a Puregene Core Kit C (QIAGEN, Valencia, CA, USA) or Oragene DNA self-collection kit (DNA Genotech, Ottawa, ON, Canada) following the manufacturer’s instructions. This study was approved by the institutional review board of Kagoshima University. All patients and family members provided informed consent for study participation. Figure 1 depicts the flowchart of our study.

### 2.2. DNA Microarray Screening

Mutation screening of 412 patients was conducted using a customized MyGeneChip^®^ CustomSeq^®^ Resequencing Array (Affymetrix, Inc., Santa Clara, CA, USA) from 2007 to 2012, targeting 28 IPN-related genes (Appendix A). The detailed methodology has been described [8].

### 2.3. NGS-Based Gene Panel Sequencing

Mutation screening has been conducted to target 60 and 72 known/candidate IPN-related genes using Illumina Miseq (Illumina Inc., San Diego, CA, USA; 435 cases) and Ion Proton (ThermoFisher Scientific, Inc., Waltham, MA, USA; 1677 cases), respectively, from 2012 and 2021. The detailed sequencing and analysis procedures have previously been described [9]. Then, in 2022, a 103-gene panel running on Ion Proton was re-designed, which added multiple new-found IPN-associated genes, amyotrophic lateral sclerosis, and distal myopathy. Until March 2022, the genetic analysis was completed in 171 cases with this new gene panel. Gene lists of all three gene panels are shown in Appendix A.

### 2.4. Whole-Exome Sequencing (WES)

Among the negative patients, 759 were further processed for WES via the Illumina Hiseq2000 platform (Illumina) or Ion Proton (ThermoFisher Scientific, Inc.). Sequencing data alignment (GRCh37/hg19) and variant calling were conducted with Burrows–Wheeler Aligner and SAMtools, or the Ion Reporter Server System. The variant files were annotated using the CLC Genomics Workbench (QIAGEN) and in-house R script. Details of our WES workflow have been previously described [10].

### 2.5. Data Analysis and Variant Interpretation

All variants obtained from the aforementioned studies were checked against the global population database (gnomAD. Available online: https://gnomad.broadinstitute.org (accessed on 1 June 2022)), Japanese control database (jMorp. Available online: https://jmorp.megabank.tohoku.ac.jp/202109/ (accessed on 1 June 2022)), and our in-house control database. All previously reported pathogenic mutations were verified by referring to the Human Gene Mutation Database Professional (Available online: https://portal.biobase-international.com/hgmd/pro (accessed on 1 June 2022)). Six in silico prediction tools were enrolled to access the variants, including SIFT/PROVEAN (Available online: http://provean.jcvi.org/index.php (accessed on 1 June 2022)), PolyPhen-2 (Available online: http://genetics.bwh.harvard.edu/pph2/ (accessed on 1 June 2022)), Mutation Assessor (Available online: http://mutationassessor.org/r3/ (accessed on 1 June 2022)), FATHMM (Available online: http://fathmm.biocompute.org.uk (accessed on 1 June 2022)), and Condel (Available online: https://bbglab.irbbarcelona.org/fannsdb/ (accessed on 1 June 2022)). All suspected disease-causing variants were validated using Sanger sequencing and interpreted according to the American College of Medical Genetics and Genomics standards and guidelines [11].

### 2.6. CNV Analysis

We analyzed the sequencing data from these patients using the Conifer or CovCopCan software to identify CNVs [3,12]. Any suspected duplication/deletion variations were then validated using MLPA (SALSA MLPA kit P405). The amplification products were analyzed on ABI PRISM 3500xl Genetic Analyzer and Coffalyser.Net software. Cases with CNVs in *PMP22* were excluded from the present study.

### 2.7. Flanking Polymerase Chain Reaction (PCR) and Repeat-Primed PCR (RP-PCR) of RFC1 Repeat Expansions

Among the 1804 undiagnosed cases, *RFC1* repeat expansion analysis was performed on 1475 cases after excluding 329 cases with autosomal dominant family history. RP-PCR was performed using specific primers for four repeat motifs, including AAAAG (benign), AAAGG (benign), AAGGG (pathogenic), and ACAGG (pathogenic), for cases without product obtained by flanking PCR [4,13]. All RP-PCR products were subjected to capillary electrophoresis using the ABI PRISM 3130xL Genetic Analyzer (Applied Biosystems, Foster City, CA, USA), and results were visualized with the Peakscanner software (Applied Biosystems). Bi-allelic *RFC1* repeat expansions were considered to fulfill the following criteria: (i) no PCR amplifiable product (<2000 bp) by flanking PCR; (ii) no smear PCR product after RP-PCR of benign repeat motif [(AAAAG)exp or (AAAGG)exp]; and (iii) presence of a decremental saw-tooth pattern on Peakscanner of RP-PCR product for pathogenic motifs, (AAGGG)exp and/or (ACAGG)exp. Primers for flanking PCR and RP-PCR are presented in Appendix A.

## 3. Results

### 3.1. Genetic Profile

Pathogenic or likely pathogenic variants in 909 (33.7%) cases were identified among the 2695 cases with suspected IPNs. *PMP22* CNVs of demyelinating CMT had been pre-excluded in this study; hence, the most common gene was *Mitofusin 2* (*MFN2*) (160, 17.6%), followed by *gap junction beta 1* (*GJB1*) (151, 16.6%), *myelin protein zero* (*MPZ*) (121, 13.3%), *membrane metalloendopeptidase* (*MME*) (35, 3.9%), *Neurofilament light polypeptide (NEFL)* (30, 3.3%), *Heat Shock Protein Family B1 (HSPB1)* (28, 3.1%), *PMP22* point mutation (22, 2.4%), *Berardinelli-Seip congenital lipodystrophy 2 (BSCL2)* (21, 2.3%), *Ganglioside Induced Differentiation Associated Protein 1 (GDAP1)* (21, 2.3%), *microrchidia family CW-type zinc finger 2 (MORC2)* (19, 2.1%), *periaxin (PRX)* (19, 2.1%), *SH3 Domain And Tetratricopeptide Repeats 2 (SH3TC2)* (19, 2.1%), *RFC1* repeat expansion (18, 2.0%), *Transthyretin (TTR)* (13, 1.4%), *Senataxin (SETX)* (12, 1.3%), *FYVE, RhoGEF And PH Domain Containing 4 (FGD4)* (10, 1.1%), *TRK-Fused Gene (TFG)* (10, 1.1%), *Dynamin 2 (DNM2)* (9, 1.0%), *Glycyl-TRNA Synthetase 1 (GARS)* (9, 1.0%), *immunoglobulin mu DNA binding protein 2 (IGHMBP2)* (9, 1.0%), *Inverted Formin 2 (INF2)* (9, 1.0%), *Superoxide Dismutase 1 (SOD1)* (9, 1.0%), *Solute Carrier Family 12 Member 6 (SLC12A6)* (9, 0.9%), etc. (Figure 2).

### 3.2. Analysis by Onset Age

The IPN cases were grouped according to the age of onset into early-onset (0–20 years), middle-onset (21–40 years), and late-onset (≥41 years). Early-onset was the most common subgroup (1230, 45.6%), followed by late-onset (914, 33.9%) and middle-onset (249, 17.4%). These findings were comparable to our previous report [9]. The highest diagnostic rate was yielded in early-onset cases (44%), and this rate tended to decline with increasing onset age (27.3% and 23.0% for middle-onset and late-onset, respectively). Then, the causative genes for each subgroup were summarized, and the most common genes were *MFN2* (early-onset; 24.4%), *GJB1* (middle-onset; 30.5%), and *MPZ* (late-onset; 19.5%). *MME* occupied the second most common causative gene in the late-onset subgroup, as well as *TTR* and *RFC1* repeat expansion Figure 3.

### 3.3. Analysis by CMT Subtypes

IPN cases were classified into demyelinating CMT (683, 25.3%), axonal CMT (1704, 63.2%), and not evoked groups (123, 4.6%; bilateral findings in 16 cases). More cases with demyelinating CMT received a genetic diagnosis than those with axonal CMT (44.9% vs. 28.0%). *GJB1* (34.9%) and *MFN2* (26.2%) were the most frequent causative genes in demyelinating and axonal CMT, respectively. Intriguingly, cases with not evoked median nerve CMAP yielded the highest diagnostic rate (61.8%), as shown in Figure 4.

### 3.4. CNV Analysis

Using Conifer or CovCopCan, together with MLPA, we identified *PMP22* CNVs in 26 cases who were excluded from the following analyses. An exon 2–6 duplication was also detected from the *MPZ* gene in one case and exon 2 deletions from *GJB1* in two cases. MLPA results of these CNVs are shown in Appendix A.

### 3.5. Bi-Allelic RFC1 Repeat Expansion

We identified bi-allelic repeat expansions in *RFC1* from 18 cases, including [(AAGGG)exp/(AAGGG)exp] (6 cases), [(ACAGG)exp/(ACAGG)exp] (6 cases), [(AAGGG)exp/(ACAGG)exp] (4 cases), [(AAGGG)exp/(AAAGG)13(AAGGG)exp] (1 case), and [(ACAGG)exp/(AAAGG)12(AAGGG)exp] (1 case). Saw-tooth patterns were observed on Peakscanner of RP-PCR product from each genotype (Appendix A) and a clinical summary of these cases are presented in Table 1 and Appendix A.

## 4. Discussion

We genetically analyzed 2695 cases with suspected IPNs using DNA microarray, multiple NGS-based gene panel sequencing systems, WES, and *RFC1* repeat expansion analysis. The overall diagnostic rate of our study was 33.7% (909 cases), and the genetic spectrum was further illustrated among varied patient subgroups with different classification strategies. This is, by far, the largest case series providing a genetic profile of patients with IPNs in Japan. Our study also revealed the *RFC1* repeat expansion prevalence from Japanese patients with IPNs.

To date, genetic spectrum studies of IPNs have been completed and updated in several countries [9,14,15,16,17,18,19,20,21,22,23,24,25,26,27,28,29]. We have reviewed these reports and compared them with our findings in Table 2. In Japan, *PMP22* CNVs testing by FISH is covered by medical insurance. A large number of patients with *PMP22* CNVs (CMT1A) were pre-excluded before referral to our laboratory; hence, any cases carrying CNVs in *PMP22* were not enrolled. *MFN2*, *GJB1*, and *MPZ* were the three main causative genes in this study, accounting for 47.5% of diagnosed IPNs. These top three genes are comparable to the findings of several other countries; however, *GJB1*, rather than *MFN2*, is the most common in their studies. This may be related to the predominance of axonal subtypes of IPNs in the current analysis, similar to Norway, but distinct from other countries, wherein demyelinating subtypes had predominance (1.1–3.18:1) [9,14,15,16,17,18,19,20,21,22,23,27,29]. The reason for this phenomenon may be associated with the *PMP22* CNV exclusion or as a regional characteristic.

Among the three subgroups with varied onset ages, late-onset IPNs yielded the lowest diagnostic rate (23.0%). *MME*, linked to late-onset axonal CMT, was the second common gene (13.8%) in this subgroup, following *MPZ*. Nevertheless, *MME* is always underestimated in clinics and is not involved in genetic screening. Additionally, we detected *TTR* mutations from 13 patients with late-onset IPNs. This gene is linked to hereditary transthyretin amyloidosis, which is one of the few treatable IPN conditions. Tafamidis and patisiran can improve the prognosis of these patients [30,31]. Collectively, genetic screening for *MME* and *TTR* genes should be performed in patients with late-onset IPNs.

We identified multi-type bi-allelic repeat expansions from 18 out of 1475 IPN cases. These genotypes were [(AAGGG)exp/(AAGGG)exp], [(AAGGG)exp/(ACAGG)exp], [(ACAGG)exp/(ACAGG)exp], [(AAGGG)exp/(AAAGG)13(AAGGG)exp], and [(ACAGG)exp/(AAAGG)12(AAGGG)exp]. Bi-allelic (AAGGG)exp in *RFC1* was found to be associated to Caucasian patients with CANVAS in 2019. Thereafter, a new pathogenic genotype, (ACAGG)exp, was reported in Japan, Indonesia, and Niue [13,32,33], and [(AAAGG)10–25(AAGGG)exp] was found only in Māori tribes [34]. Analysis that targets (ACAGG)exp should be added in the Asia-Pacific region, unlike other regions. While in Japan, [(AAAGG)10–25(AAGGG)exp] should be considered.

Most *RFC1*-related disorders are adult-onset. However, three cases in our study were with early-onset at age 10 or younger. Therefore, analyzing *RFC1* repeat expansion is necessary regardless of onset age. All of these cases developed muscle weakness or atrophy, and most of them had sensory disturbance or subclinical sensory neuropathy. Cerebellar ataxia and vestibular dysfunction are the core symptoms of CANVAS that were observed in only five (5/16, 31.3%) patients. The most common phenotype in our study is sensory-motor neuropathy. Several patients had chronic cough (3/7, 42.9%), autonomic neuropathy (5/15, 33.3%), muscle cramps (4/9, 44.4%), and hyperCKemia (7/13, 53.8%). *RFC1*-related neuropathy is often accompanied by chronic cough and autonomic disturbance [6,35], which are suggestive signs of *RFC1*-related disorders among IPNs. To date, hyperCKemia and muscle cramps have been reported only in patients carrying (ACAGG)exp [13,33]. However, our study found it as common among various genotypes. Therefore, hyperCKemia and muscle cramps should not be considered as characteristic signs for (ACAGG)exp, and every repeat motif should be analyzed.

The strength of our study includes the comprehensive genetic analyses, which demonstrated the genetic diversity and proportion data of IPNs in Japan. An overall understanding with respect to the genetic IPN spectrum would help us make more rational and efficient genetic testing strategies, including gene panel analysis, WES, CNVs analysis, and *RFC1* analysis. Given the ongoing cost reduction of WES and the development of sequencing technologies, such as long-read sequencing, the aforementioned testing strategies would be optimized. Overall, the findings in this study would facilitate the early diagnosis of patients with IPNs and provide patients with the best care to improve their disease prognosis.

## Figures and Tables

**Figure 1 biomedicines-10-01546-f001:**
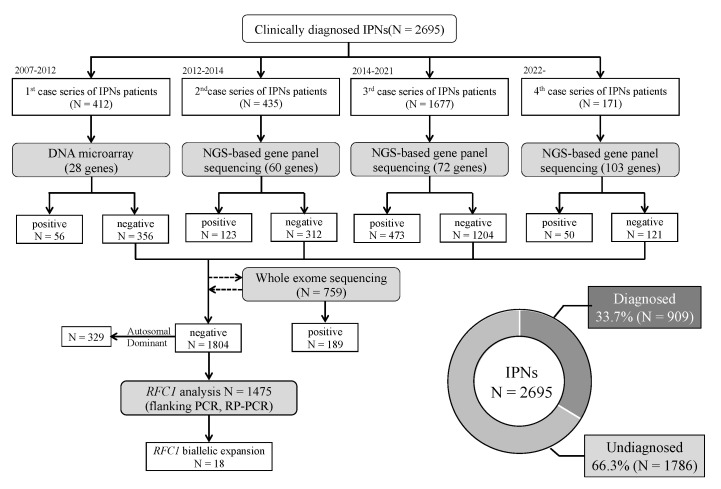
Flowchart of comprehensive genetic analyses of IPNs in our study. DNA microarray, NGS-based gene panel sequencing, and whole-exome sequencing were used to analyze 2695 cases with clinically diagnosed IPNs. Among the undiagnosed cases, *RFC1* analysis was performed on 1475 cases, which identified bi-allelic *RFC1* repeat expansions in 18 cases.

**Figure 2 biomedicines-10-01546-f002:**
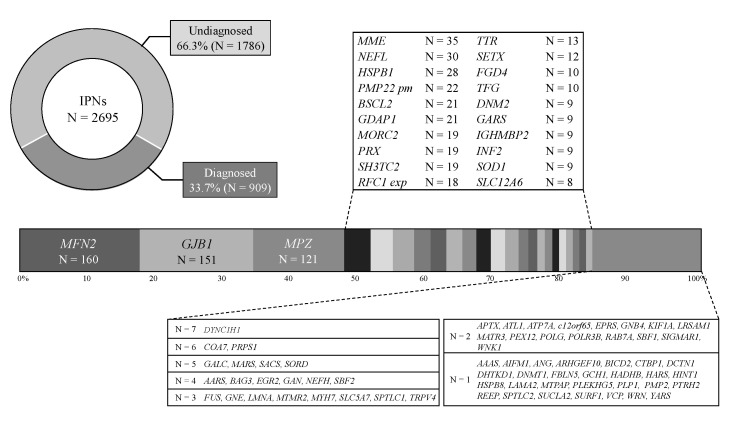
Genetic spectrum of IPNs in our Japanese case series. Pathogenic or likely pathogenic variants were identified in 909 (33.7%) cases among 2695 cases with suspected IPNs. pm: point mutation; exp: repeat expansion.

**Figure 3 biomedicines-10-01546-f003:**
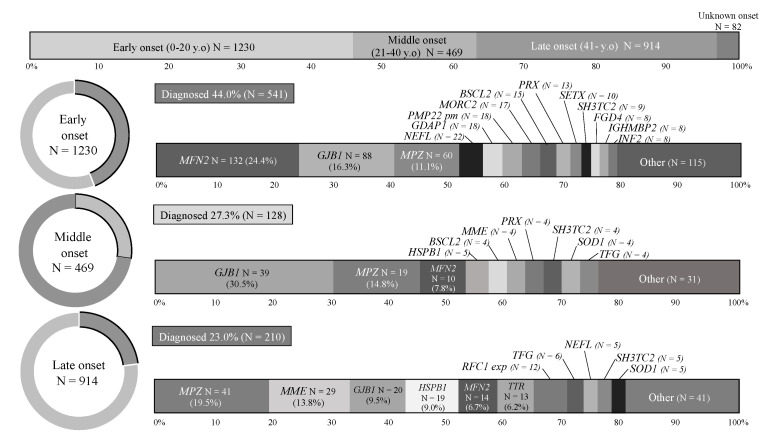
Genetic spectrum of each IPN subgroup with different onset ages. Diagnostic rates and genetic spectrums of IPNs with early-onset (0–20 years, *n* = 1230), middle-onset (21–40 years, *n* = 469), and late-onset (>41 years, *n* = 914). pm: point mutation; exp: repeat expansion.

**Figure 4 biomedicines-10-01546-f004:**
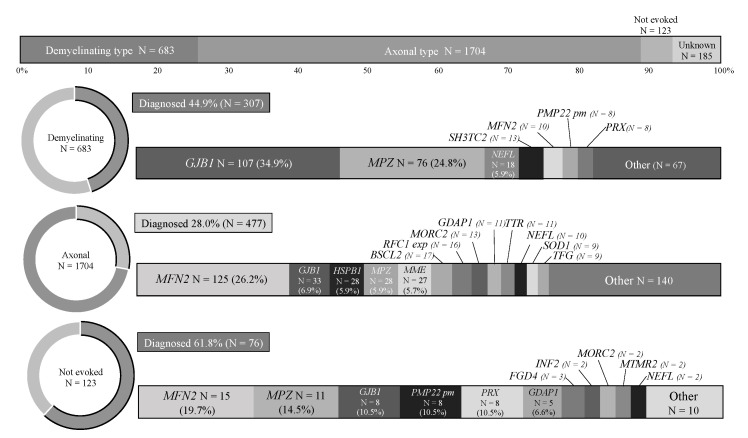
Diagnostic rate and genetic spectrum of IPNs with demyelinating, axonal, and not evoked subtypes. pm: point mutation.

**Table 1 biomedicines-10-01546-t001:** Clinical manifestations and electrophysiological findings in 18 cases with *RFC1* repeat expansions.

*RFC1* Repeat	(AAGGG)exp	(ACAGG)exp	(AAGGG)exp	(AAGGG)exp	(ACAGG)exp	All Cases
	(AAGGG)exp	(ACAGG)exp	(ACAGG)exp	(AAAGG)13 (AAGGG)exp	(AAAGG)12 (AAGGG)exp	
Patient number	N = 6	N = 4	N = 6	N = 1	N = 1	N = 18
Onset age	28.8 ± 22.8	49.75 ± 27.6	64.7 ± 11.6	25	50	46.3 ± 24.2
Sex	Male 6, Female 0	Male 4, Female 0	Male 4, Female 2	Male	Female	Male 15, Female 3
Muscle weakness	6/6 [100%]	4/4 [100%]	5/6 [83.3%]	+	+	17/18 [94.4%]
Muscle atrophy	3/4 [75%]	2/3 [66.7%]	5/6 [83.3%]	+	+	12/15 [80%]
Hyporeflexia	5/5 [100%]	3/4 [75%]	6/6 [100%]	−	+	15/17 [88/2%]
Sensory disturbance	5/5 [100%]	4/4 [100%]	6/6 [100%]	−	+	16/17 [94.1%]
Cerebellar ataxia	1/4 [25%]	1/4 [25%]	2/3 [66.7%]	−	+	5/16 [31.3%]
Cerebellar atrophy	1/3 [33.3%]	0/2 [0%]	2/3, 66.7%	−	−	3/10 [30%]
Vestibular dysfunction	0/3 [0%]	1/4 [25%]	0/3 [0%]	NA	+	2/11 [9.1%]
Chronic cough	0/1 [0%]	1/3 [33.3%]	2/3 [66.7%]	NA	NA	3/7 [42.9%]
Pyramidal sign	0/5 [0%]	1/4 [25%]	0/6 [0%]	−	−	1/17 [5.9%]
Parkinsonism	1/4 [25%]	0/4 [0%]	0/6 [0%]	−	−	1/16 [6.3%]
Cognitive impairment	0/2 [0%]	1/3 [33.3%]	0/5 [0%]	−	−	1/12 [8.3%]
Involuntary movement	0/2 [0%]	0/4 [0%]	1/6 [16.7%]	−	−	1/14 [7.1%]
Autonomic dysfunction	1/3 [33.3%]	2/4 [50%]	1/6 [16.7%]	+	−	5/15 [33.3%]
Muscle cramp	1/1 [100%]	1/3 [33.3%]	2/4 [50%]	NA	−	4/9 [44.4%]
Hyper CKemia	3/3 [100%]	1/3 [33.3%]	2/5 [40%]	+	−	7/13 [53.8%]
Median MNCV (m/s)	54.2 ± 6.1	47.5 ± 1.3	53.6 ± 5.8	50.6	56.5	52.5 ± 6.1
Median CMAP (mV)	10.8 ± 3.9	4.5 ± 4.2	6.2 ± 2.7	6.2	8.48	8.0 ± 4.1
Median SCV (m/s)	44.5 ± 12.6	31	39.8	48	58	45.8 ± 11.3
Median SNAP (μV)	23.8 ±25.3	0.7 ± 1.1	2.6 ± 5.7	0.4	2.6	9.1 ± 15.6
Tibial MNCV (m/s)	43.9 ± 2.2	37.5 ± 2.4	39.0 ± 6.4	40.2	43.8	40.9 ± 4.5
Tibial CMAP (mV)	7.8 ± 6.3	2.5 ± 2.9	4.8 ± 4.1	3.2	6.15	5.6 ± 5.0
Sural SCV (m/s)	50.4 ± 5.9	NA	NA	NE	NE	50.4 ± 5.9
Sural SNAP (μV)	8.3± 4.5	0	0	NE	NE	3.0 ± 4.8

CK: creatine kinase; CMAP: compound motor action potential; MNCV: motor nerve conduction velocity; SNAP: sensory nerve action potential; SCV: sensory nerve conduction velocity; NE: not evolved; NA: not available. Normal range: median CMAP > 3.1 mV; median MCV > 49.6 m/s; median SNAP > 7.0 μV; median SCV > 47.2 m/s; tibial CMAP > 4.4 mV; tibial MCV > 41.7 m/s; sural SNAP > 5.0 μV; sural SCV > 40.8 m/s.

**Table 2 biomedicines-10-01546-t002:** Comparison of current and previous studies on diagnostic rate and genetic proportion of common IPN-related genes.

	*PMP22*CNVs	*MFN2*	*GJB1*	*MPZ*	*MME*	*NEFL*	*HSPB1*	*PMP22 pm*	*BSCL2*	*GDAP1*	*MORC2*	*PRX*	*SH3TC2*	*RFC1* exp	*TTR*	DiagnosticRate (%)	Demyelinating Type: Axonal Type
Our study (N = 2695)	*1	160 [5.9%]	151 [5.6%]	121 [4.5%]	35 [1.3%]	30 [1.1%]	28 [1.0%]	22 [0.8%]	21 [0.8%]	21 [0.8%]	19 [0.7%]	19 [0.7%]	19 [0.7%]	18 [0.7%]	13 [0.5%]	33.7	1:2.49
Japan, 2011 (N = 354) [14]	53 [15.0%]	14 [4.0%]	25 [7.1%]	25 [7.1%]	/	8 [2.3%]	0 [0%]	10 [2.8%]	/	1 [0.3%]	/	5 [1.4%]	/	/	/	40.3	1.79:1
Japan, 2018 (N = 1005) [9]	*1	66 [6.6%]	66 [6.6%]	51 [5.1%]	8 [0.8%]	9 [0.9%]	14 [1.4%]	13 [1.3%]	6 [0.6%]	8 [0.8%]	/	4 [0.4%]	5 [0.5%]	/	/	30.0	1:2.42
Korea, 2016 (N-78) [15]	15 [19.2%]	1 [1.3%]	9 [11.5%]	2 [2.6%]	/	0 [0%]	0 [0%]	1 [1.3%]	0 [0%]	0 [0%]	/	0 [0%]	1 [1.3%]	/	/	21.8	1.1:1
China, 2019 (N = 150) [16]	52 [34.7%]	9 [7%]	19 [14%]	[3%] *2	/	/	/	1% *2	/	1% *2	/	/	/	/	/	66.7	1.59:1
China, 2021 (N = 435) [17]	99 [22.8%]	[10.1%] *2	[13.5%] *2	[5.0%] *2	[0.5%] *2	[0.7%] *2	[0.9%] *2	[3.2%] *2	0 [0%]	[2.1%] *2	/	/	[2.1%] *2	/	/	70	1.22:1
Taiwan, 2019 (N = 427) [18]	208 [48.7%]	14 [3.3%]	40 [9.4%]	14 [3.3%]	/	8 [1.9%]	2 [0.5%]	4 [0.9%]	2 [0.5%]	2 [0.5%]	1 [0.2%]	0 [0%]	3 [0.7%]	/	0 [0%]	73.1	2.8:1
UK, 2012 (N = 1607) [19]	415 [25.8%]	60 [3.7%]	147 [9.1%]	31 [1.9%]	/	4 [0.2%]	3 [0.2%]	11 [0.7%]	2 [0.1%]	12 [0.7%]	/	/	9 [0.6%]	/	/	44.3	1.52:1
Germany, 2013 (N = 776) [20]	180 [23.2%]	12 [1.5%]	47 [6.1%]	21 [2.7%]	/	0 [0%]	/	8 [1.0%]	/	0 [0%]	/	0 [0%]	0 [0%]	/	/	58	2.35:1
Norway, 2013 (N = 435) [21]	20 [4.6%] *3	11 [2.5%]	12 [2.8%]	10 [2.3%]	/	2 [0.5%]	/	0 [0%]	/	/	/	/	/	/	/	16.6	1:2.08
Spain, 2013 (N-438) [22]	184 [42.0%]	4 [0.9%]	56 [12.8%]	19 [4.3%]	/	4 [0.9%]	7 [1.6%]	2 [0.5%]	/	42 [9.6%]	/	4 [0.9%]	28 [6.4%]	/	/	83.3	1.69:1
Italy, 2014 (N = 197) [23]	100 [50.8%]	2 [1.0%]	14 [7.1%]	7 [3.6%]	/	1 [0.5%]	1 [0.5%]	7 [3.6%]	/	8 [4.1%]	/	/	3 [1.5%]	/	/	75.1	1.98:1
Italy, 2019 (N = 566) [24]	233 [41.2%]	8 [1.4%]	33 [5.8%]	36 [6.4%]	/	1 [0.2%]	14 [2.5%]	7 [1.2%]	8 [1.4%]	4 [0.7%]	/	/	0 [0%]	/	/	62.2	/
Denmark, 2019 (N = 1442) [25]	236 [16.4%]	24 [1.7%]	32 [2.2%]	27 [1.9%]	/	1 [0.07%]	0 [0%]	3 [0.2%]	0 [0%]	0 [0%]	/	0 [0%]	1 [0.07%]	/	/	23.1	/
USA, 2014 (N = 17,880) [26]	1823 [10.2%]	138 [0.8%]	215 [1.2%]	170 [1.0%]	/	22 [0.1%]	10 [0.06%]	30 [0.2%]	/	22 [0.1%]	/	1 [0.005%]	26 [0.1%]	/	/	18.5	/
Brazil, 2021 (N = 503) [27]	116 [23.1%]	12 [2.4%]	23 [4.6%]	6 [1.2%]	0 [0%]	3 [0.6%]	0 [0%]	6 [1.2%]	0 [0%]	7 [1.4%]	/	2 [0.4%]	3 [0.6%]	/	/	78.3	3.18:1
Turkey, 2021 (N = 55) [28]	0 [0%]	1 [1.8%]	2 [3.6%]	0 [0%]	/	0 [0%]	0 [0%]	0 [0%]	0 [0%]	2 [3.6%]	/	1 [1.8%]	2 [3.6%]	/	/	23.6	/
Turkey, 2022 (N-649) [29]	151 [23.3%]	15 [2.3%]	47 [7.2%]	6 [0.9%]	/	2 [0.3%]	2 [0.3%]	3 [0.5%]	/	18 [2.8%]	/	6 [0.9%]	17 [2.6%]	/	/	46.5	2.47:1

*1. *PMP22* CNVs were excluded from the analysis. *2. Percentages were described, but the numbers of cases were not. *3. For some cases, CMT1A have been excluded by other laboratories.

## Data Availability

The data supporting the conclusions of this article will be made available by the authors, without undue reservation.

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
