# Peer review of "Comprehensive Genetic Analyses of Inherited Peripheral Neuropathies in Japan: Making Early Diagnosis Possible"

_biomedicines, 2022, doi:10.3390/biomedicines10071546_

Round 1

Reviewer 1 Report

Interesting paper.

According to the heterogeneity of the published studies, it might be wise to indicate in the title « in Japan ». Also, it is difficult to really know which exact disorders are included in the terminology « inherited peripheral neuropathies » ; in fact of course it is mainly CMT. So, I would suggest to replace in the title « inherited peripheral neuropathies » by CMT.

Other few inherited neuropathies than CMT which are mentioned in this text, such as amyloidosis and RFC1 repeated expansions could be maintained in the results by indicating that the techniques which have been used have also identified these types of neuropathy.

Concerning the data analysis and the variant interpretation, the discussion is only technical ; in routine practice it is quite often necessary to consider combinations of the clinical phenotype with hereditary transmission, genetic, bioinformatic, electrophysiological data and sometimes nerve pathology ; functional evidence may be also necessary to assess the pathogenicity of genetic variants in CMT patients.

So, is it possible to know how many variants of uncertain significance have been discussed ?

It seems that the authors have used 2 different panels : one which screens 72 genes and since recently, one which screens 103 genes. Do they plan to test the negative cases of the first panel using the second panel ?

Author Response

DATE 23, Jun, 2022

Dear Reviewer,

We are grateful for your careful reading of our manuscript and for giving insightful comments. In response to all your comments, a point-by-point response letter has been prepared, and the manuscript has been modified correspondingly.

We look forward to the publication of our manuscript on Biomedicines.

Our responses to your comments are as follows:

(Reviewer’s comment)

According to the heterogeneity of the published studies, it might be wise to indicate in the title « in Japan ». Also, it is difficult to really know which exact disorders are included in the terminology « inherited peripheral neuropathies » ; in fact of course it is mainly CMT. So, I would suggest to replace in the title « inherited peripheral neuropathies » by CMT.

--------------------

(Our response)

We agree with the reviewer that the patients enrolled in this study are predominantly CMT. However, there are still patients clinically diagnosed with other types of IPN, like HSAN. Moreover, multiple genes involved in our gene panels are not CMT disease-causing genes, such as SOD1, TTR, and HSAN-related genes. Taken together, it would be inaccurate and possibly misleading to replace "IPN" with "CMT" in the title. So we decided to keep "inherited peripheral neuropathies” in our title. And since our study is limited to Japanese, we added "in Japan" to the title.

--------------------

(Revisions to our manuscript)

[Title]

Comprehensive Genetic Analyses of Charcot-Marie-Tooth disease in Japan: Making Early Diagnosis Possible

------------------------------------------------------------------------------------------------------------

(Reviewer’s comment)

Other few inherited neuropathies than CMT which are mentioned in this text, such as amyloidosis and RFC1 repeated expansions could be maintained in the results by indicating that the techniques which have been used have also identified these types of neuropathy.

Concerning the data analysis and the variant interpretation, the discussion is only technical ; in routine practice it is quite often necessary to consider combinations of the clinical phenotype with hereditary transmission, genetic, bioinformatic, electrophysiological data and sometimes nerve pathology ; functional evidence may be also necessary to assess the pathogenicity of genetic variants in CMT patients.

So, is it possible to know how many variants of uncertain significance have been discussed ?

------------------------------------------------------------------------------------------------------------ (Our response)

We thank the reviewer for this suggestion. On the basis of the ACMG guidelines, the number of variants of uncertain significance is extremely large. For example, there are plenty of cases carrying one to three variants of uncertain significance detected by CMT103 panel sequencings. And exome analysis reveals much more variants classified as uncertain significance. Therefore, it is difficult for us to involve the variants classified as uncertain significance in this paper.

------------------------------------------------------------------------------------------------------------

(Reviewer’s comment)

It seems that the authors have used 2 different panels : one which screens 72 genes and since recently, one which screens 103 genes. Do they plan to test the negative cases of the first panel using the second panel ?

------------------------------------------------------------------------------------------------------------ (Our response)

We did not apply the updated CMT103 gene panel to reanalyze the negative cases, who were originally screened by 72-gene panel analysis. Instead, we have been performing whole exome sequencings for these undiagnosed cases after gene panel sequencing.

Reviewer 2 Report

This is a very interesting and well-written paper about genetic diagnosis of suspected inherited peripheral neuropathy (IPN) in a large series of Japanese patients. This report provides very interesting insight on the topic with new data on the performance of genetic testing in these patients. I find the data about RFC1 expansion in this series are of great value. Overall, the paper deserves being published provided some points are clarified.

Major point:

One of the problems of IPN is the definition of cases. Indeed, IPN might be suspected because of a clear family history (meaning at least two members have been examined by clinical/electrodiagnostic means), because of a suspected family history (meaning the affected relatives are unavailable), or because of a clinical phenotype which is highly suggestive of IPN. In the present study, it would be very helpful to know what the authors mean by “clinically diagnosed with IPN means”? Was family history positive for all patients or were there patients with a clinically suggestive phenotype but no family history? What was the proportion of each type of presentation?

Minor point:

Concerning the patients with no conduction study recordable from the median nerve (not evoked group): was a motor nerve conduction study performed on both sides? Additionally, was a MNCV elicitable from the ulnar nerve? What was the result if any?

From a general point of view, it would be interesting to have some insight about the evolving strategy for genetic testing in patients with suspected IPN in Japan or at least in this center. Do the authors think a first approach with a mini-panel followed by systematic exome sequencing would be a better strategy? Indeed, performing large panel sequencing from the beginning is time consuming and the majority of elucidated cases have sequence variation in about only 10 different genes.

Author Response

DATE 23, Jun, 2022

Dear Reviewer,

We appreciate your careful review of our manuscript and for giving insightful comments. In response to your comments, a point-by-point response letter has been prepared, and the manuscript has been modified correspondingly.

We look forward to the publication of our manuscript on Biomedicines.

Our responses to your comments are as follows:

(Reviewer’s comment)

This is a very interesting and well-written paper about genetic diagnosis of suspected inherited peripheral neuropathy (IPN) in a large series of Japanese patients. This report provides very interesting insight on the topic with new data on the performance of genetic testing in these patients. I find the data about RFC1 expansion in this series are of great value. Overall, the paper deserves being published provided some points are clarified.

Major point:

One of the problems of IPN is the definition of cases. Indeed, IPN might be suspected because of a clear family history (meaning at least two members have been examined by clinical/electrodiagnostic means), because of a suspected family history (meaning the affected relatives are unavailable), or because of a clinical phenotype which is highly suggestive of IPN. In the present study, it would be very helpful to know what the authors mean by “clinically diagnosed with IPN means”? Was family history positive for all patients or were there patients with a clinically suggestive phenotype but no family history? What was the proportion of each type of presentation?

--------------------

(Our response)

We thank the reviewer for this comment. The DNA samples and clinical notes were collected throughout Japan. The diagnosis of IPN were made by experienced neurologists/pediatricians at local clinics/hospitals, based on clinical symptoms and family history. Regarding the inheritance pattern of all 2695 cases, 1698 cases were sporadic, and 954 cases had a positive family history or consanguinity; 45 cases with unavailable family history data.We have added the information to the manuscript.

--------------------

(Revisions to our manuscript)

2.1. Patients

We collected 2,695 unrelated Japanese IPNs patients from neurological and neuropediatric departments throughout Japan from 2007 to March 2022. The clinical diagnosis of IPN was made by each neurologist or pediatric neurologist. Among these patients, positive family history or consanguinity were found from 954 cases, and 1698 cases were sporadic; 45 cases had no available family history data.

------------------------------------------------------------------------------------------------------------

(Reviewer’s comment)

Minor point:

Concerning the patients with no conduction study recordable from the median nerve (not evoked group): was a motor nerve conduction study performed on both sides? Additionally, was a MNCV elicitable from the ulnar nerve? What was the result if any?

------------------------------------------------------------------------------------------------------------ (Our response)

We thank the reviewer for this comment. We reviewed the clinical notes of 123 patients whose median nerve was classified as not evoked group. Therein, from 16 patients, the NCS study of median nerves were performed bilaterally, in which 15 of them were not evoked on either side, and one patient had a markedly decreased CMAP (0.8 mV) on the contralateral side.

NCS records of ulnar nerve were only available in 35 patients, and they were presumed to be demyelinating (MNCV < 38 m/s; 20 cases) and axonal (MNCV >38 m/s; 15 cases). (Data not shown)

3.3. Analysis by CMT subtypes

IPN cases were classified into demyelinating CMT (683, 25.3%), axonal CMT (1,704, 63.2%), and not evoked groups (123, 4.6%; bilateral findings in 16 cases).

------------------------------------------------------------------------------------------------------------

(Reviewer’s comment)

From a general point of view, it would be interesting to have some insight about the evolving strategy for genetic testing in patients with suspected IPN in Japan or at least in this center. Do the authors think a first approach with a mini-panel followed by systematic exome sequencing would be a better strategy? Indeed, performing large panel sequencing from the beginning is time consuming and the majority of elucidated cases have sequence variation in about only 10 different genes.

------------------------------------------------------------------------------------------------------------ (Our response)

We thank the reviewer for these insightful comments. Currently, considering the cost and effort of whole exome sequencing, and the facilities we have in-house, we still prefer to conduct the large panel sequencing first. The combination of mini-panel and exome sequencing could be a great strategy, which we would definitely take into account to optimize our screening workflow in the future. We have added discussion in our manuscript.

--------------------

(Revisions to our manuscript)

The strength of our study includes the comprehensive genetic analyses, which demonstrated the genetic diversity and proportion data of IPNs in Japan. An overall understanding with respect to the genetic IPN spectrum would help us make more rational and efficient genetic testing strategies, including gene panel analysis, WES, CNVs analysis, and RFC1 analysis. Given the ongoing cost reduction of WES and the development of sequencing technologies, like long-read sequencing, the aforementioned testing strategies would be optimized. Overall, the findings in this study would facilitate the early diagnosis of patients with IPNs and provide patients with the best care to improve their disease prognosis.
